# Immunoglobulin Replacement Therapy is critical and cost-effective in increasing life expectancy and quality of life in patients suffering from Common Variable Immunodeficiency Disorders (CVID): A health-economic assessment

**Philippe van Wilder[1], Irina Odnoletkova[1], Mehdi Mouline[1], Esther de Vries**[2,3] *

1 Research Centre in Health Economics, School of Public Health, Université Libre de Bruxelles, Brussels, Belgium, 2 Dept Tranzo, Tilburg School of Social and Behavioral Sciences, Tilburg University, Tilburg, The Netherlands, 3 Laboratory of Medical Microbiology and Immunology, Elisabeth-Tweesteden Hospital, Tilburg, The Netherlands

* e.devries@tilburguniversity.edu

## Abstract

### Background

Common variable immunodeficiency disorders (CVID), the most common form of primary antibody deficiency, are rare conditions associated with considerable morbidity and mortality. The clinical benefit of immunoglobulin replacement therapy (IgGRT) is substantial: timely treatment with appropriate doses significantly reduces mortality and the incidence of CVID-complications such as major infections and bronchiectasis. Unfortunately, CVID-patients still face a median diagnostic delay of 4 years. Their disease burden, expressed in annual loss of disability-adjusted life years, is 3-fold higher than in the general population. Hurdles to treatment access and reimbursement by healthcare payers may exist because the value of IgGRT is poorly documented. This paper aims to demonstrate cost-effectiveness and cost-utility (on life expectancy and quality) of IgGRT in CVID.

### Methods and findings

With input from a literature search, we built a health-economic model for cost-effectiveness and cost-utility assessment of IgGRT in CVID. We compared a mean literature-based dose (≥450mg/kg/4wks) to a zero-or-low dose (0 to ≤100 mg/kg/4wks) in a simulated cohort of adult patients from time of diagnosis until death; we also estimated the economic impact of diagnostic delay in this simulated cohort.

Compared to no or minimal treatment, IgGRT showed an incremental benefit of 17 life-years (LYs) and 11 quality-adjusted life-years (QALYs), resulting in an incremental cost-effectiveness ratio (ICER) of €29,296/LY and €46,717/QALY. These results were robust in a sensitivity analysis. Reducing diagnostic delay by 4 years provided an incremental benefit of

**Data Availability Statement:** All relevant data are within the manuscript and its Supporting Information files.

**Funding:** Funding for this project was provided by the Plasma Protein Therapeutics Association, PPTA Europe, Boulevard Brand Whitlock 114/b4, 1200 Brussels, Belgium. The PPTA staff provided useful comments regarding the study design and during the manuscript preparation. However, the authors performed the work and took decisions regarding the work independently of the funders. The funders had no role in the final study design, data collection and analysis, decision to publish, or preparation of the final version of the manuscript.

**Competing interests:** I have read the journal's policy and the authors of this manuscript have the following competing interests: both EdV and PvW received funding from PPTA to perform the work. EdV has received a research grant from Shire/Takeda for performing independent research on unclassified antibody deficiency. This does not alter our adherence to PLOS ONE policies on sharing data and materials.

six LYs and four QALYs compared to simulated patients with delayed IgGRT initiation, resulting in an ICER of €30,374/LY and €47,495/QALY.

## Conclusions

The health-economic model suggests that early initiation of IgGRT compared to no or delayed IgGRT is highly cost-effective. CVID-patients' access to IgGRT should be facilitated, not only because of proven clinical efficacy, but also due to the now demonstrated cost-effectiveness.

## Introduction

A rare disease is often characterized by high morbidity and mortality [1] and is defined as a condition with a low number of affected patients (typically <5/10,000) by regulatory authorities [2]. Due to their rarity, public awareness regarding these diseases is very low or absent. Even within the medical community, knowledge about clinical, diagnostic, and therapeutic approaches in rare diseases is often limited and patients may suffer from misdiagnosis or undertreatment.

The incidence of common variable immunodeficiency disorders (CVID) in the population is estimated at 1: 25,000. CVID constitutes a heterogeneous group of immune defects characterized by hypogammaglobulinemia and failure of specific antibody production resulting in poor responses to vaccination and increased susceptibility to mainly respiratory infections such as chronic sinusitis, chronic otitis media, bronchitis, and pneumonia. It equally affects men and women and is associated with increased morbidity and mortality [3]. CVID is also characterized by an array of other comorbidities [4,5]. Complications of CVID can be divided into structural damage to the lungs due to severe and/or recurrent infections resulting in bronchiectasis, and damage to organs as consequences of immune dysregulation. Autoimmune, inflammatory and lymphoproliferative conditions have a 5-fold higher prevalence in CVID [Janssen et al., submitted] compared to the general population. Although the link between CVID and increased rates of infection has been known for decades, the evolving understanding of comorbidities related to the immune dysregulation is more recent [6–11]. The burden of a disease can be expressed in Disability-adjusted Life-years (DALYs), which are defined as the sum of Years-of-life-lost (YLL) and Years-lived-with-disability (YLDs). Higher DALYs indicate a greater burden of disease. The annual health loss attributable to CVID is estimated at 36,785 DALYs per 100,000 patients, 3.5 times higher than the 10,510 DALYs per 100,000 persons that are annually lost in the general population [3].

Immunoglobulin replacement therapy (IgGRT) is the standard of care in CVID [12]. IgGRT is highly effective in reducing the mortality rate and the incidence of major infections in CVID-patients. The meta-analysis of Orange et al. [13] showed that the incidence of pneumonia declined by 27% with each 100 mg/dL increment in trough serum IgG-levels. There was a linear relationship between the trough serum IgG-level and the IgGRT-dose: trough serum IgG-level increased by 121 mg/dL with each increase of administered IgGRT-dose by 100 mg/kg. However, analysis of the European Society for Immunodeficiencies (ESID) Registry database, containing data on 2,700 European patients with CVID, identified that current practice shows a significant variation in monthly IgGRT-doses across European countries with a mean (standard deviation; SD) of 454 (196) mg/kg; the analysis also showed that a diagnostic delay of several years occurred in over 80% of the cohort [3]. Each year of diagnostic delay

increased the relative mortality risk by 3.8%. These findings represent a significant public health concern. Early-initiated IgGRT may improve patients' health and quality of life and may also reduce overall healthcare and societal costs by avoiding or reducing the incidence of longer-term complications related to primary immunodeficiency (PID) [14].

Health-economic assessments consider value-for-money; they link the health effects (or 'incremental clinical benefit') of an intervention (or 'health technology') to its additional cost. These assessments consider all costs and effects associated with a disease and the proposed health technologies in a given period. For a chronic disease, a lifetime horizon will often be adopted in such an assessment [15]. The value of a new health technology can be expressed as an incremental cost-effectiveness ratio (ICER), which represents the costs per Life-year (LY) gained, or incremental cost-utility ratio (ICUR), which represents the costs per Quality-adjusted Life-year (QALY) gained. These commonly used measures estimate the cost-effectiveness (ICER) and cost-utility (ICUR) of a new technology compared to existing alternatives; simply speaking, the smaller the ICER and ICUR of a new technology are, the more efficient it is.

Health resources are scarce; only technologies that are both clinically effective *and* cost-effective represent the most value for society. Some healthcare authorities will use ICER-thresholds to facilitate the decision to reimburse a new technology. The National Institute for Health and Care Excellence (NICE) [16] uses a threshold value of £20–30,000 per QALY below which a technology is considered cost-effective. Technologies having a high ICER compared to the alternatives may be subject to refusal by healthcare authorities because of a lack of cost-effectiveness, even if clinically effective. This position is not always in the patients' best interests; therefore, it is important to provide healthcare authorities with assessments that will lead to optimal decisions safeguarding patient health. This approach is especially important for rare diseases, because low patient numbers will limit the return on investment. It is reasonable to propose a higher ICER threshold for technologies when used in rare diseases. In CVID, the cost-effectiveness of IgGRT is poorly investigated. As a result, decision makers and clinicians make coverage and treatment decisions based solely on *cost estimations* rather than on the *cost-effectiveness assessment* of IgGRT.

This paper aims to demonstrate the value of IgGRT as long-term therapy for CVID patients through the development of a health-economic assessment framework. To our knowledge, there are no health-economic assessments that compare mean literature-based-dose IgGRT to zero-or-low-dose IgGRT or investigate the health-economic consequences of diagnostic delay in CVID patients. However, to apply such assessments to technologies used for rare diseases can be a challenge: rare disease frequency may negatively affect the level of the available evidence, and substantial uncertainty may remain about the cost-effectiveness estimates [17]. This paper describes a health-economic experiment: the development of *a core model* in which the use of mean literature-based-dose IgGRT is compared to zero-or-low-dose IgGRT in simulated CVID-patients. This health-economic model estimates the impact of IgGRT and of diagnostic delay in terms of life-long benefits and costs for CVID.

## Materials and methods

We used the ESID Registry [3] and the 2016 International Consensus on Common Variable Immunodeficiency Disorders (ICON) [18] to define the CVID patient population. To obtain input for the health-economic model, we searched the literature relating to the natural history of CVID up to June 2018, in Embase, Cochrane Library, DARE and PubMed. Additional literature searching was continued until model finalization (August 2019). Based on the relevant data from the studies identified, we built a CVID health-economic model, with consideration

of the modeling Task Force recommendations of the International Society for Pharmacoeconomics and Outcomes Report (ISPOR) [19].

In the base-case scenario of our health-economic experiment, we started with a cohort of 2,000 just-diagnosed *simulated* CVID-patients all aged 30 years, and assigned these to either:

1. an 'intervention' group of 1,000 simulated patients treated with a mean literature-based-dose IgGRT ($\geq$450 mg/kg/4 weeks),

2. a 'control' group of 1,000 simulated patients treated with zero-or-low-dose IgGRT regimens (0 to $\leq$100 mg/kg/4 weeks).

The impact of diagnostic delay was estimated by simulating:

3. a 'delayed intervention' group; we achieved this by replacing the early cycles of the intervention group by the values of the control group (e.g. if the diagnostic delay is set at 4 years, the values of the 1-month cycles 1 to 48 of the intervention group are then replaced by the values of the 1-month cycles 1 to 48 of the control group).

The groups were modeled over a time horizon of 50 years. Age, time horizon and the dose of $\geq$450mg/kg/4 weeks were selected based on the mean age at diagnosis, the median life expectancy on IgGRT and the mean IgG-dose administered in the ESID Registry study, respectively [3]; the limit of $\leq$100mg/kg/4 weeks was selected because this was the lowest dose in a meta-analysis of clinical studies that evaluated the incidence of pneumonia based on IgGRT-dose and IgG trough levels in antibody deficient patients [13].

This base-case analysis assessed the effects (using LYs and QALYs) and costs from a healthcare payer perspective; other economic aspects (e.g. ability to work and pay taxes) were not taken into account. Mortality was derived from the Medical Research Council study of 1969 [20] and the Cunningham-Rundles study of 1999 [21] to provide estimates for the simulated control and intervention groups respectively. The mortality data from the original publications were adjusted by comparing the mortality in the general population of the two study periods: to correct and update the old mortality estimates, we reduced the CVID treatment specific mortality rates (MR) by comparing the mortality rates of the general population (gen pop) in the distinct time periods:

$$\text{new MR}_{\text{CVID}} = \text{old published MR}_{\text{CVID}} \times (\text{recent MR}_{\text{gen pop}}/\text{old MR}_{\text{gen pop}})$$

Due to the scarcity of appropriate quality of life studies, QALYs were not estimated by relevant utility (valued quality of life; U) measurements; instead, its counterpart *dis*utility (decrement in utility; D) was estimated from Global Burden of Disease (GBD) studies [22] for each health state to which a CVID can transition during a lifetime (see below and Fig 1). For composite health states, we calculated disutility as the result of the different single comorbidity disutility values weighted by the comorbidity incidences among CVID-patients. For example, for 'Autoimmune disease' a disability weight was used based on disutility linked to thyroid disease, Crohn's disease, irritable bowel syndrome, anemia and thrombocytopenic purpura; for 'Cancer', a mix of early and advanced cancer states was used; for patients who could belong to more than one health state (e.g. having cancer and suffering from a minor infection), the worst disutility value was used. Utility–a quality of life adjusted health state–was then calculated as U = 1—D. An overview of all utility estimates is given in S1 Table.

The costs of CVID from a healthcare perspective (e.g. diagnosis, healthcare use, disease and co-morbidity treatments) were obtained from published cost estimates per health state. Because this is a comparative assessment, we focused on costs of IgGRT acquisition/administration and the costs of health states with different transition probabilities (from one health

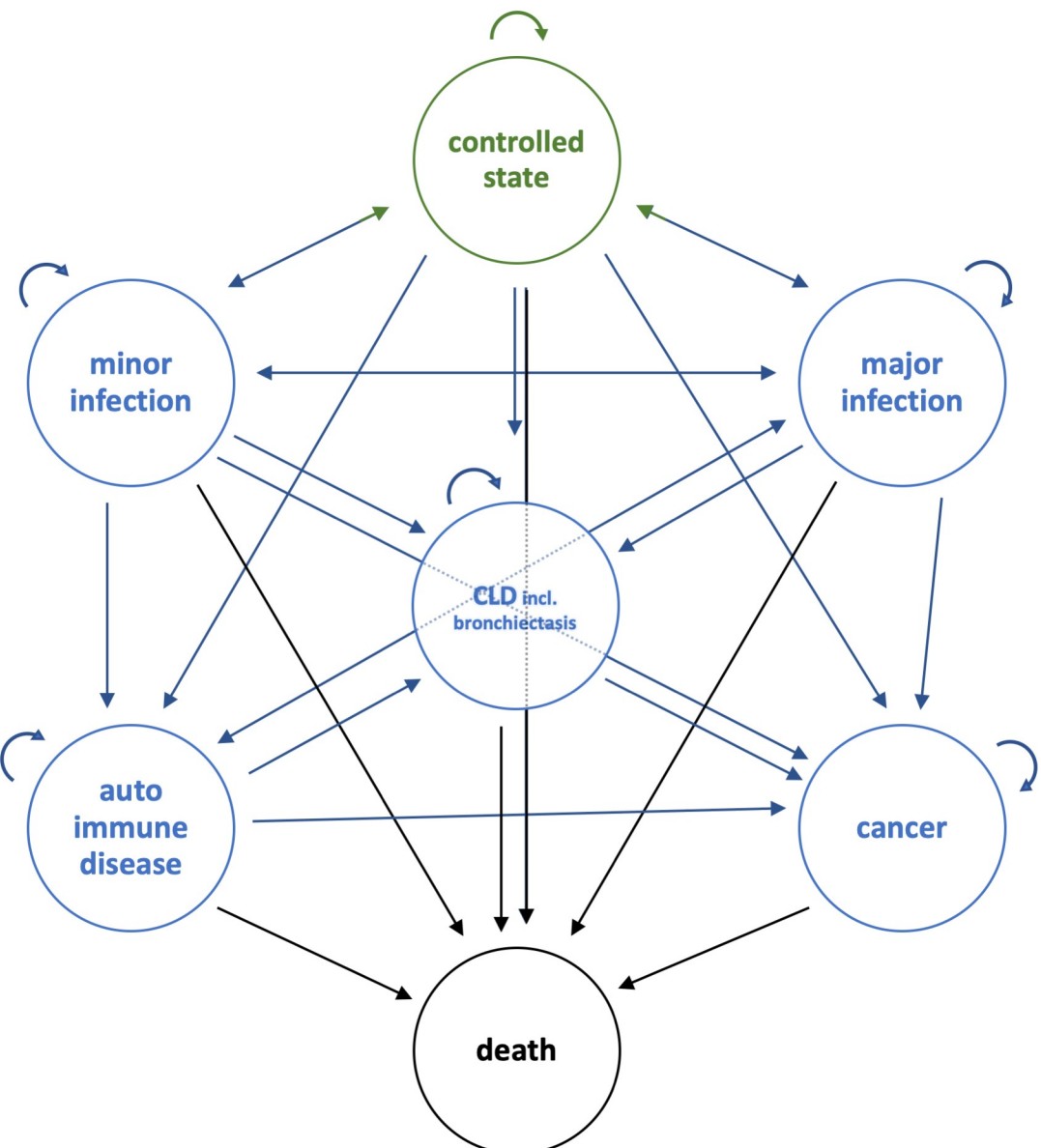

**Fig 1. Markov chain model with seven CVID health states.** CLD, chronic lung disease (including bronchiectasis).

state to another) between the intervention and the control group. These transition probabilities were derived from the literature search [3,18,23]. An overview of the relevant information sources for the cost estimates is provided in S2 Table. Probabilities (p) were converted to rates (r) [24] and transformed back to probabilities for use in the economic model, by using the following equations (t = time):

$$\text{Rate} = \ln (1-p)/t \quad \text{Probability} = 1-\exp (-rt)$$

The base-case transition probabilities can be found in S3 Table.

We evaluated the impact of a variation of each relevant model parameter using one-way sensitivity analyses. The uncertainty around the parametric assumptions of the model input

data was tested through a probabilistic sensitivity analysis. Specific distributions were assigned to transition probabilities (β-distribution), utilities (β-distribution) and costs (γ-distribution) (S4 Table). A Monte Carlo simulation with 1,000 runs for each group was done, by randomly sampling the values within the bounds of the reported 95% confidence intervals (CIs) of the point estimates.

In the base-case scenario, future costs and QALYs were estimated to decrease in value by different percentages for effects (1,5%) and costs (3%), in line with the WHO guide for immunization programs [25], because we expect treatment effects to occur early and to affect the course of longer-term disease progression. To run the base-case analyses, percentages of 1.5% for effects and 3% for costs per annum, were used per Belgian guidelines [26] Sensitivity analyses included estimated decrease in value between 0% and 5% as proposed by the EUnetHTA-guidelines [27]. Specific beta distributions were linked to transition probabilities and to utilities, gamma distributions have been selected for cost elements. The model assumptions and the assessment results were evaluated by experts in clinical immunology and compared to published findings.

The data used in the health economic model were managed in two software programs. Basic data manipulations, e.g., converting probabilities to rates, in Microsoft Excel 2016; advanced data management, Boolean logic, algorithm for Monte Carlo simulations, and programming documentation in IBM SPSS Statistics 26. The base-case ICER-computations were done in parallel in both software programs to check the calculation accuracy.

## Results

### Design of the health-economic model

The literature search confirmed the limited evidence level of clinical research and the lack of well-designed health-economic modeling studies in CVID. We used all relevant, good quality data that could be retrieved from the identified studies, including studies on the CVID disease registries in the US and in Europe with follow-up data on large cohorts (>1,000 patients) as well as smaller multi-annual observational studies. Based on the input from the literature search, a Markov chain cohort model with seven health states was designed (Fig 1). The model simulates progression for a diagnosed CVID-patient from a 'controlled state' to health states with various infectious diseases (minor infections, major infections), chronic lung disease (CLD, including bronchiectasis), autoimmune disease, cancer and death. Minor infections include common airway infections such as bronchitis, sinusitis and otitis. Major infections are predominantly pneumonia and bacterial meningitis. We assumed that from the controlled state and the infectious health states, transitions are possible to any other health state. There is no reversal to controlled state if a patient enters the bronchiectasis health state or worse; from bronchiectasis (CLD), only cancer and death are possible. From cancer, only death is modelled as a possible transition.

The model follows the simulated cohorts of 1,000 CVID-patients each (intervention group, control group, delayed intervention group, see Methods) from the time of diagnosis at age 30 over a time horizon of 50 years. Considering the specified health states, cycles of 1-month duration were used. Based on the ESID Registry study [3] and clinical expertise, each 1,000-patient cohort was distributed at the start to be in the defined health states as follows: 10 controlled state, 540 minor infection, 25 major infection, 200 autoimmunity, 200 bronchiectasis, 25 cancer, 0 dead. We assumed the effect of IgGRT results in lower transition probabilities for mortality, the occurrence of infections and development of bronchiectasis (see S4 Table); we also assumed there would be no IgGRT effect on the occurrence of autoimmune diseases or cancer.

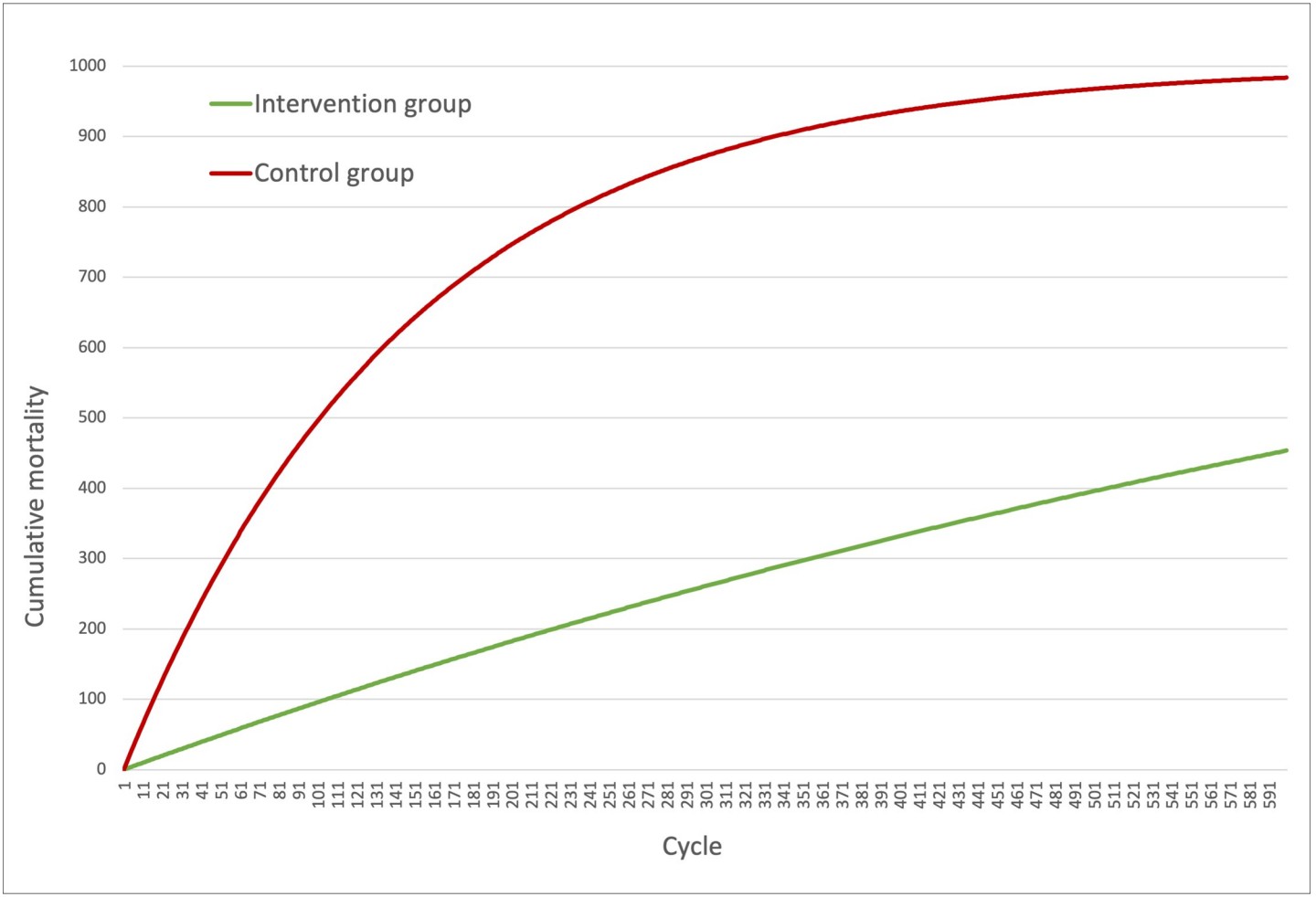

**Fig 2. Markov trace on mortality in the control and the intervention groups.** X-axis = cycle, from first to last (600). Y-axis = cumulative number of deaths. Dead_IgG = cumulative number of deaths in the intervention group (mean literature-based-dose IgGRT). Dead_Comp = cumulative number of deaths in the control group (zero-to-low-dose IgGRT).

## Health-economic assessment results

The simulation results suggest IgGRT has a significant positive impact on <u>mortality</u> in the intervention group compared with the control group. The intervention group reached a cumulative mortality of 45% compared with 98% in the control group. A similar result is seen in the

**Table 1. Health effects in the base-case scenario.**

|  | Control group | Intervention group | Difference |
|---|---|---|---|
|  | *(IgGRT 0 to ≤ 100mg/kg/4 weeks)* | *(IgGRT ≥ 450mg/kg/4 weeks)* |  |
| **Total health care costs** | € 54,519.44* | € 562,877.18* | **€ 508,357.74** |
| **LYs gained** | 10.30 | 27.66 | **17.35** |
| **QALYs gained** | 6.86 | 17.75 | **10.88** |

The clinical benefit of IgGRT has an efficiency outcome of €29,296 per LY and €46,717 per QALY.

* The healthcare cost is limited in the control group as compared to the intervention group because of the high mortality (with no financial healthcare cost) and the lack of IgGRT costs. The opposite is true for the intervention group: IgGRT increases survival but has an associated substantial pharmaceutical cost (acquisition and administration).

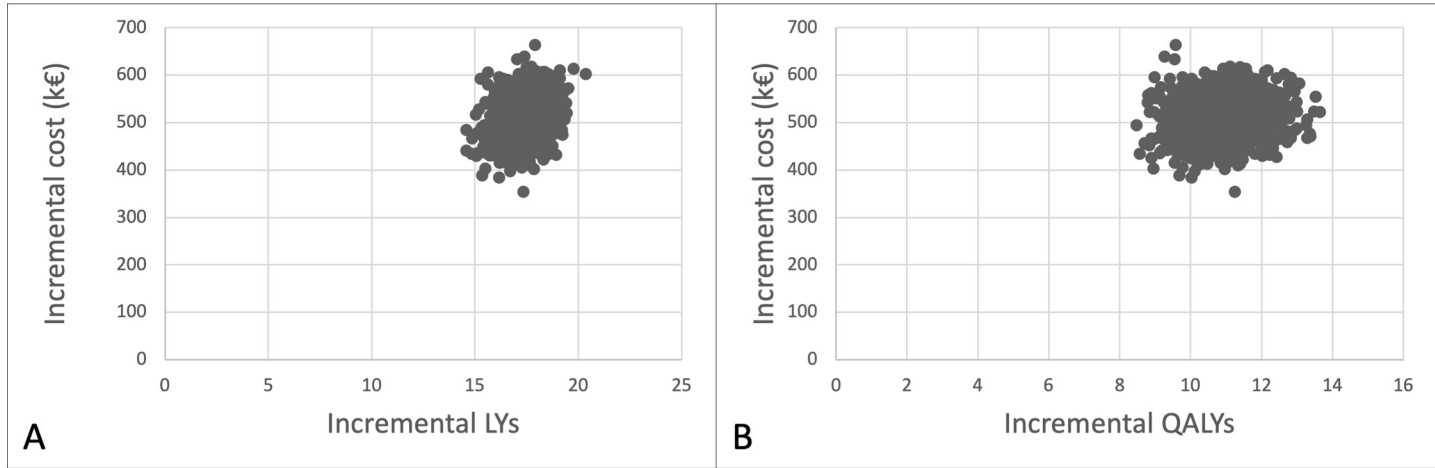

**Fig 3. Cost-effectiveness planes for IgGRT.** A. Costs versus LYs. B. Costs versus QALYs.

overall median <u>survival</u>, which was 102 months in the control group, whereas a median overall survival was not reached in the intervention group (Fig 2).

The <u>direct healthcare costs</u> and the <u>benefits in LYs and QALYs</u> are shown in Table 1 for the control and intervention groups when decreasing the value of future health effects at 1.5% and costs at 3% (see Methods). The incremental cost of IgGRT, spread over 50 years, is associated with an incremental benefit of 10.9 QALYs and 17.4 LYs per patient.

We investigated the impact of varying the input parameters on the cost-effectiveness outcomes. One-way parameter changes had a modest impact on the ICER (€/LY) and ICUR (€/QALY) (S5 Table), further illustrating the cost-effectiveness of IgGRT in CVID (S5 Table). The multivariate sensitivity results were obtained by a Monte Carlo simulation in which relevant transition probabilities, utilities and costs were allowed to vary according to their estimated distribution. The simulation indicates the robustness of the results as illustrated in Fig 3. This figure provides the cost-effectiveness planes for respectively LYs and QALYs. Both cost-effectiveness planes are composed of the simulated estimates of costs and effects in the Monte Carlo runs, which are all positioned in a well delimited cloud of 1,000 dots. From the cost-effectiveness plane it can be deduced that the use of IgGRT is cost-effective in 80% of cases if the threshold is set at €31,676 per LY or €48,359 per QALY. IgGRT is cost-effective in 90% of cases if the threshold is set at €32,762 per LY or €53,992 per QALY.

The <u>health impact of diagnostic delay</u> and subsequently delayed initiation of IgGRT in CVID had a deleterious health impact: 1.7–11.8 LYs were lost in the model with diagnostic delay tested in a range of 1–10 years, with a mean loss of 4.0 QALYs and 6.2 LYs for a

**Table 2. Health effects in the 4-year diagnostic delay scenario.**

| | Delayed intervention group | Intervention group | Difference |
|---|---|---|---|
| | *(IgGRT ≥ 450mg/kg/4 weeks after a 4-year diagnostic delay)* | *(IgGRT ≥ 450mg/kg/4 weeks)* | |
| **Total health care costs** | € 374,171.02 | € 562,877.18 | **€ 188,706.16** |
| **LYs gained** | 21.44 | 27.66 | **6.21** |
| **QALYs gained** | 13.77 | 17.75 | **3.97** |

Treating early is cost-effective with efficiency estimates of € 30,373.55/LY and € 47,494.63/QALY

The distribution at start across the distinct health states for the delayed intervention group is identical to the distribution in the intervention group.

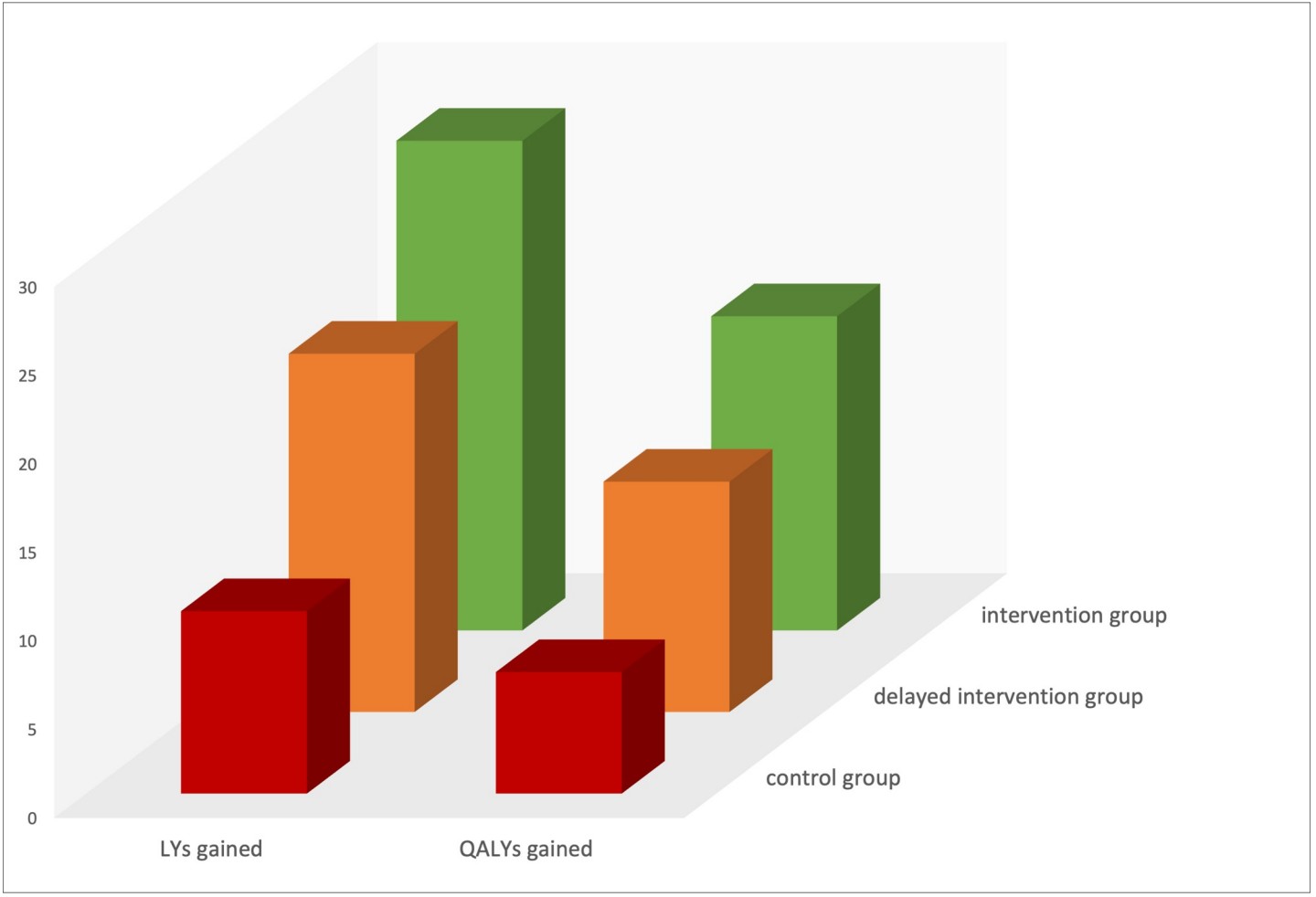

**Fig 4. The difference in health outcome between the three simulated cohorts.** Intervention group: 1,000 simulated patients treated with a mean literature-based-dose of IgGRT (≥450 mg/kg/4 weeks); delayed intervention group (4 years): The first 48-months cycles of the intervention group were replaced by the values of the control group; control group: 1,000 simulated patients treated with zero-to-low-dose IgGRT regimens (0 to ≤100 mg/kg/4 weeks). Y-axis: Years gained. LYs, Life-years; QALYs, Quality-adjusted Life-years.

diagnostic delay of 4 years (Table 2). The difference in health outcome between the three simulated cohorts is impressive, as is clearly illustrated by Fig 4. And importantly, treating early is still cost-effective, with efficiency estimates of €30,373.55/LY and €47,494.63/QALY.

## Discussion

### Main messages

In our health-economic assessment model, we show that IgGRT is cost-effective in CVID, a serious disease known to negatively affect a patient's life expectancy and quality of life [3]. It is also a rare disease (1: 25,000), with inherent low or absent public awareness and limited knowledge in the medical community about clinical, diagnostic, and therapeutic approaches, leading to misdiagnosis and undertreatment.

The use of IgGRT is the current standard of practice in CVID, as replacement therapy to increase a patient's antibody levels. Timely treatment with appropriate doses significantly

reduces mortality and the incidence of infectious complications and bronchiectasis [13]. IgGRT has demonstrated clinical effectiveness but its *efficiency* (cost-effectiveness) is poorly documented, creating a potential hurdle to treatment access and to reimbursement by health-care payers.

In a health-economic assessment model, the ICER shows how much clinical benefit is gained at how much extra cost. ICER-thresholds are a tool for disentangling cost-effective from non-cost-effective technologies. The WHO considers health technologies with an ICER below or equal to the gross domestic product per inhabitant (GDP; total value of goods and services produced within a country) as 'highly cost-effective', technologies with an ICER below 3x the GDP as 'cost-effective' technologies [28]. This WHO cost-effectiveness-threshold formulation ensures it is applicable in various geographic regions with differences in economic productivity. A plausible estimate of the ICER for IgGRT is needed to facilitate patient access to immunoglobulin replacement therapy.

This paper calculates this estimate in its health-economic assessment of IgGRT in CVID. The results demonstrate that the cost of IgGRT is associated with an incremental *clinical* benefit including 17 extra LYs and 11 extra QALYs. The base-case ICER for mean literature-based-dose IgGRT as compared to zero-or-low-dose IgGRT was close to €30,000/LY and €47,000/QALY. IgGRT costs vary across geographical regions (see S2 Table). However, the economic evaluation results were modestly impacted by changes in treatment costs in the model. The obtained ICERs were mainly based on data from European and US studies and were distinctly below regular WHO cost-effectiveness-thresholds in the Western world (3x GDP would exceed €100,000 in the EU and the US). Applying the health-economic model to developing countries would require replacing our input data by appropriate local data (as yet not sufficiently available); the cost-effectiveness results would have to be compared to the local ICER-threshold, expressed in local GDP.

The cost-effectiveness of IgGRT we found in our model for CVID is *impressive* because ICERs for treatments for rare diseases are often considerably higher than the standard cost-effectiveness thresholds [29]. IgGRT in CVID even yields ICER-estimates that are of the same magnitude as those for the–by far–more common influenza vaccination program for adults [30].

The model also demonstrated that treatment delay reduces the LYs and QALYs of CVID-patients: a 4-year treatment delay had a tremendous impact in our study, resulting in a loss of 4 QALYs and 6 LYs. Repeating the analyses for different diagnostic delay periods resulted in comparable ICERs close to €31.000 per LY. Early diagnosis with timely treatment was a highly cost-effective approach compared to treatment delay. Cost-effectiveness and clinical benefit should encourage early treatment and there certainly is a *serious ethical concern* regarding diagnostic and treatment delay.

## Limitations

The study has several limitations. The main limitation relates to the lack of accurate, up-to-date data in the CVID-literature on the incidence and sequence of comorbidity events and on the nature of risk factors, a well-known caveat for evaluating treatments in rare diseases [31]. Detailed multi-national clinical follow-up studies are needed to solve this problem, but these are complicated to perform. Therefore, we had to derive utilities from disutilities, and not–as usual–from quality of life measurements in CVID-patients. Disease proxies had to be used for some distinct health states. Unadjusted cost estimates of significant co-morbidities came from various published sources; it would be important to do a comparison to real-world economic impact data. One structural limitation relates to the distinct nature of health states; e.g. if

autoimmunity co-exists with infection, the patient utility may be affected differently than when it is the only health state for that patient, even if we selected the worst health state. We assumed that the hazard rate in mortality is constant. A recent report regarding health-economic assessments on the use of immunoglobulins across an array of distinct diseases, including CVID, encountered similar study limitations [32], confirming the lack of good quality data.

We used a Markov chain cohort model with seven distinct health states with infectious as well as autoimmune and malignant comorbidities. Different economic models could be proposed; the ISPOR good research practice guideline expresses the difficulty related to the choice of appropriate model design [19]. The general rule is that the model design and the number of health states should reflect the natural history of disease as much as possible, without inflating the mathematical and conceptual complexity [29]. We have tried to match this rule as much as possible using a literature search and evaluation by clinical experts in the field. We did not include distinct age subgroups, higher doses of IgGRT (e.g. in case of bronchiectasis) or other medicinal products that could be used for the treatment of patients with CVID. These could be subjects for later studies.

We had to estimate the mortality rates in different health states for the intervention and control groups in the model. The results are affected by the applied percentages for reduced value of future health states; the worst—but still cost-effective—results are obtained when using the same percentages for health effects and costs. We chose not to use this worst-case scenario because the health impact of IgGRT on CVID is already observed early and affects longer-term progression. In this clinical situation, WHO advocates against the use of the same percentage for health effects and costs [25] (this assumption is also reflected in the Belgian and Dutch economic guidelines [27]).

The model could underestimate the cost-effectiveness of IgGRT. Another study, which included CVID and other PIDs with a higher impact of IgGRT, indicated that IgGRT may even be cost-saving [14]. It is prudent to suggest that adopting the cost estimates of Modell et al. [14] would even make IgGRT the dominant therapy, i.e. offering more health benefits at a lower cost than using no or delayed treatment.

## Conclusion

CVID is a chronic, progressive disease in which the antibody deficiency has an *early and sustained impact* on the incidence of infections and mortality. It has been demonstrated that IgGRT has a marked impact on ameliorating comorbidities caused by the antibody deficiency. Before IgGRT, CVID-mortality was 70% 12 years after diagnosis [20]; that is no longer the case today. Still, greater awareness leading to early diagnosis is needed to avoid unnecessary loss of healthy LYs.

This study performed a simulated economic evaluation of CVID-treatment with IgGRT. The model showed an average benefit of 17 LYs and 11 QALYs for mean literature-based-dose IgGRT, resulting in an ICER of €29,296/LY and €46,717/QALY. These estimates are comparable with those for current influenza vaccination programs. Treating CVID-patients promptly was also very cost-effective in the model: an average benefit of 6 LYs and 4 QALYs, resulting in an ICER of €30,374/LY and €47,495/QALY for a 4-year reduction in diagnostic delay. These cost-effectiveness and cost-utility results are *far below the WHO-thresholds of efficiency*. These findings should be a strong incentive for competent authorities and medical and scientific associations to actively promote awareness, emphasizing the importance of early diagnosis, as well as to facilitate patient access to IgGRT, supporting timely treatment of CVID. To facilitate

this, authorities should also encourage plasma donation for the production of IgGRT for patients with CVID and other PIDs.

In conclusion, the results illustrate both the clinical effectiveness as well as the cost-effectiveness of IgGRT in CVID. Based on guidelines and best practices, early, physician directed IgGRT should be encouraged by competent authorities and healthcare payers alike.

## Supporting information

**S1 Table. Disutility values and utility calculation.**
(PDF)

**S2 Table. Data sources for economic modeling and evaluation.**
(PDF)

**S3 Table. Matrix of transition probabilities between the different health states.**
(PDF)

**S4 Table. Monte Carlo simulations.**
(PDF)

**S5 Table. Results from one-way sensitivity analyses.**
(PDF)

## Acknowledgments

We are grateful to the PPTA staff for providing useful comments regarding the study design and during the manuscript preparation.

## Author Contributions

**Conceptualization:** Philippe van Wilder, Irina Odnoletkova.

**Data curation:** Philippe van Wilder, Esther de Vries.

**Formal analysis:** Philippe van Wilder, Mehdi Mouline.

**Funding acquisition:** Irina Odnoletkova.

**Methodology:** Philippe van Wilder, Irina Odnoletkova, Esther de Vries.

**Supervision:** Esther de Vries.

**Validation:** Philippe van Wilder.

**Visualization:** Philippe van Wilder, Esther de Vries.

**Writing – original draft:** Philippe van Wilder, Esther de Vries.

**Writing – review & editing:** Philippe van Wilder, Irina Odnoletkova, Mehdi Mouline, Esther de Vries.

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
