## [Decision Letter · Decision Letter 0]

24 Nov 2020

PONE-D-20-23871

Immunoglobulin Replacement Therapy is critical and cost-effective in increasing life expectancy and quality of life in patients suffering from Common Variable Immunodeficiency Disorders (CVID): A health-economic assessment

PLOS ONE

Dear Dr. de Vries,

Thank you for submitting your manuscript to PLOS ONE. After careful consideration, we feel that it has merit but does not fully meet PLOS ONE’s publication criteria as it currently stands. Therefore, we invite you to submit a revised version of the manuscript that addresses the points raised during the review process.

We look forward to receiving your revised manuscript.

Kind regards,

Kednapa Thavorn, PhD

Academic Editor

PLOS ONE

Journal Requirements:

2.Thank you for stating the following in the Competing Interests section:

[I have read the journal's policy and the authors of this manuscript have the following competing interests: both EdV and PvW received funding from PPTA to perform the work. EdV has received a research grant from Shire/Takeda for performing independent research on unclassified antibody deficiency.].

3. We note you have included a table to which you do not refer in the text of your manuscript. Please ensure that you refer to Table 2 in your text; if accepted, production will need this reference to link the reader to the Table.

Reviewers' comments:

Reviewer's Responses to Questions

**Comments to the Author**

1. Is the manuscript technically sound, and do the data support the conclusions?

Reviewer #1: Yes

Reviewer #2: Yes

2. Has the statistical analysis been performed appropriately and rigorously? 

Reviewer #1: Yes

Reviewer #2: Yes

3. Have the authors made all data underlying the findings in their manuscript fully available?

Reviewer #1: Yes

Reviewer #2: Yes

4. Is the manuscript presented in an intelligible fashion and written in standard English?

Reviewer #1: Yes

Reviewer #2: Yes

5. Review Comments to the Author

Reviewer #1: This manuscript is very good writing. Here are my concern and comments as follow;

1. I was wondering why you chose 5 years for study delayed in diagnostic? Because you mentioned that CVID-patients would face a median diagnostic delay of 4 years.

2. The mortality was derived from the studies those are quite old. I was wondering if there is the mortality from recent study available that would be better.

3.Reference No. 16 Line 434

"Accessed June 9th, ... via"Please specify the year you accessed the reference.

Reviewer #2: This is a well written paper that aims to provide new information in an under-researched area: cost-effectiveness and cost-utility of immunoglobulin replacement therapy in patients with common variable immunodefiency disorder. Minor revision only is recommended.

Abstract:

Line 44: ‘imput’ should be spelt ‘input.’

General comment on mathematical model vs economic model. Given this paper is about cost utility/cost effectiveness, please use health economic model and delete ‘mathematical’ model. It is confusing to the reader, as it appears they are different models, but it’s just an inconsistency in language.

Methods/Results:

Line 189: please provide justification for the assignment of distributions selected (gamma, beta etc).

It is unclear what software was used to undertake the modelling, please state.

It is unclear if the same patient numbers started in each health state for the delayed intervention model. Please clarify.

The analysis was undertaken from a health care system perspective, but it is unclear which healthcare system (ie: the country), please state.

It is unclear if any adjustments were made to costs (inflation or conversion), please provide this information in Table S2 and/or S4.

6. PLOS authors have the option to publish the peer review history of their article (what does this mean?). If published, this will include your full peer review and any attached files.

Reviewer #1: No

Reviewer #2: No

---

## [Author Response · Author response to Decision Letter 0]

7 Dec 2020

Journal Requirements:

1. Please ensure that your manuscript meets PLOS ONE's style requirements, including those for file naming. We have checked our manuscript and adapted it according to the ‘manuscript body formatting guidelines’ and the ‘title, author, affiliations formatting guidelines’.

2.Thank you for stating the following in the Competing Interests section: [I have read the journal's policy and the authors of this manuscript have the following competing interests: both EdV and PvW received funding from PPTA to perform the work. EdV has received a research grant from Shire/Takeda for performing independent research on unclassified antibody deficiency.]. "This does not alter our adherence to PLOS ONE policies on sharing data and materials.” Please include your updated Competing Interests statement in your cover letter. We have complied with these requests.

3. We note you have included a table to which you do not refer in the text of your manuscript. Please ensure that you refer to Table 2 in your text; if accepted, production will need this reference to link the reader to the Table. We have corrected this omission.

Reviewer #1: 

This manuscript is very good writing. We thank the reviewer for his/her positive evaluation of our manuscript.

Here are my concern and comments as follow;

1. I was wondering why you chose 5 years for study delayed in diagnostic? Because you mentioned that CVID-patients would face a median diagnostic delay of 4 years. 

We are grateful to the reviewer to mention this apparent inconsistency. In the source that we used for estimating the diagnostic delay (ESID Registry study (2018)), it is indeed mentioned that the median delay was 4 years. Accordingly, in the text we corrected and replaced all the results obtained for a delay of 5 years by the results obtained for a delay of 4 years. The conclusion may remain as formulated because, as we mentioned in the manuscript, the ICER/ICUR was fairly robust when we changed the duration of the diagnostic delay in a univariate sensitivity analysis.

2. The mortality was derived from the studies those are quite old. I was wondering if there is the mortality from recent study available that would be better. 

The mortality data for the control group (< 100mg/kg of IgG) were derived from the MRC-study which was indeed published more than 4 decades ago; in our literature study we did not find a more recent study describing mortality for untreated CVID-patients. We assume that the widespread use of IgG became standard practice for CVID-patients in the years after the MRC-study, making more recent randomized trials with an ‘untreated’ control group unethical.

For the IgGRT-group we used the Cunningham-Rundles study published in 1999; more recently, the ESID Registry study (2018) also provided estimates of mortality on IgGRT (with a lower mortality risk than in the Cunningham study). In our manuscript we have chosen to keep the Cunningham reference and to use the ESID-data in a sensitivity analysis. 

To correct and update the old mortality estimates, we reduced the CVID treatment specific mortality rates (MR) by comparing the mortality rates of the general population (gen pop) in the distinct time periods: 

new MRCVID = old published MRCVID x (recent MR gen pop / old MR gen pop)

We have added this last sentence to the manuscript.

3.Reference No. 16 Line 434 "Accessed June 9th, ... via"Please specify the year you accessed the reference. 

The year we accessed reference 16 was 2020: we corrected the reference accordingly. 

Reviewer #2:

This is a well written paper that aims to provide new information in an under-researched area: cost-effectiveness and cost-utility of immunoglobulin replacement therapy in patients with common variable immunodefiency disorder. Minor revision only is recommended. We thank the reviewer for his/her positive evaluation of our manuscript.

Abstract:

Line 44: ‘imput’ should be spelt ‘input.’ We have corrected this. (NB: because of other adjustments, the line numbers have changed somewhat.)

General comment on mathematical model vs economic model. Given this paper is about cost utility/cost effectiveness, please use health economic model and delete ‘mathematical’ model. It is confusing to the reader, as it appears they are different models, but it’s just an inconsistency in language. We have complied with this request.

Methods/Results:

Line 189: please provide justification for the assignment of distributions selected (gamma, beta etc). 

The specific distributions linked to transition probabilities, to cost elements and to utilities have been selected based on the modeling methods described in: Richard Edlin, Christopher McCabe, Claire Hulme, Peter Hall, Judy Wright. Cost Effectiveness Modelling for Health Technology Assessment. A Practical Course. Adis (2014) ISBN 978-3-319-15743-6 ISBN 978-3-319-15744-3 (eBook) DOI 10.1007/978-3-319-15744-3. This document is available in the public domain and we have now provided it in S1 Appendix.

We used bèta distributions for transition probabilities, as proposed in chapter six, gamma distributions for cost estimates, as described in chapter seven, and bèta distributions for utilities (or disutility if utility should encompass 0), see chapter seven. We have included this information in the Methods section.

It is unclear what software was used to undertake the modelling, please state. 

The data used in the health economic model were managed in 2 software programs. Basic data manipulations, e.g., converting probabilities to rates, in Microsoft Excel 2016; advanced data management, Boolean logic, algorithm for Monte Carlo simulations, and programming documentation in IBM SPSS Statistics 26. The base-case ICER-computations were done in parallel in both software programs to check the calculation accuracy. We have included this information in the Methods section.

It is unclear if the same patient numbers started in each health state for the delayed intervention model. Please clarify. 

The distribution at start across the distinct health states for the delayed intervention group is identical

to the distribution in the other control groups; we did not find relevant data to decide otherwise. We have added this information to Table 2.

We assume this is a conservative assumption because more patients may be in a more severe health state due to the delay. Thus, our results might underestimate the value of IgGRT.

The analysis was undertaken from a health care system perspective, but it is unclear which healthcare system (ie: the country), please state. 

The study is based on data from the literature, originating in many different countries. We used estimates of direct medical costs from the literature, mostly from various countries in the Western European region but also from other geographic regions. Thus, the supporting data mainly originate from countries with a well-developed public healthcare funding in the healthcare system.

This is mentioned in the Discussion section.

It is unclear if any adjustments were made to costs (inflation or conversion), please provide this information in Table S2 and/or S4. 

The cost estimates came from various geographic sources. Considering the regional diversity in cost estimates, we built a core health economic model, in which the published cost data were used and converted to actual value in Euro, rounded to the upper 100 Euro level. We believe more accurate estimates per health state are needed in country-specific analyses for country-specific use; but a country-specific health economic analysis would require checking the country-specific databases and tariffs which goes beyond the purpose of our study. We compensated the uncertainty in our estimates by performing sensitivity analyses on the efficiency impact of changing the cost estimates of: 

- the immunoglobulin G replacement therapy

- the health state ‘major infections’

- the health state ‘auto-immune disease’

- the health state ‘chronic lung disease’

- the health state ‘cancer’

In our analyses, the cost parameter estimates were varied as follows:

Results from Monte Carlo simulations on cost estimates

 IgGRT Infection Major Auto-immune disease Chronic Lung Disease Cancer

Mean 1.663 8.991 329 453 600

Median 1.656 8.976 328 452 599

Minimum 1.065 6.520 234 303 448

Maximum 2.227 12.079 453 584 818

We believe the range used in our estimates should compensate for the relative lack of precision of the cost estimates per health states, due to geographical variation.

This information was added to S4 Table.

---

## [Decision Letter · Decision Letter 1]

17 Feb 2021

Immunoglobulin Replacement Therapy is critical and cost-effective in increasing life expectancy and quality of life in patients suffering from Common Variable Immunodeficiency Disorders (CVID): A health-economic assessment

PONE-D-20-23871R1

Dear Dr. de Vries,

We’re pleased to inform you that your manuscript has been judged scientifically suitable for publication and will be formally accepted for publication once it meets all outstanding technical requirements.

Kind regards,

Kevin Lu, PhD

Academic Editor

PLOS ONE

Additional Editor Comments (optional):

Reviewers' comments:

Reviewer's Responses to Questions

**Comments to the Author**

1. If the authors have adequately addressed your comments raised in a previous round of review and you feel that this manuscript is now acceptable for publication, you may indicate that here to bypass the “Comments to the Author” section, enter your conflict of interest statement in the “Confidential to Editor” section, and submit your "Accept" recommendation.

Reviewer #1: All comments have been addressed

2. Is the manuscript technically sound, and do the data support the conclusions?

Reviewer #1: Yes

3. Has the statistical analysis been performed appropriately and rigorously? 

Reviewer #1: Yes

4. Have the authors made all data underlying the findings in their manuscript fully available?

Reviewer #1: Yes

5. Is the manuscript presented in an intelligible fashion and written in standard English?

Reviewer #1: Yes

6. Review Comments to the Author

Reviewer #1: (No Response)

7. PLOS authors have the option to publish the peer review history of their article (what does this mean?). If published, this will include your full peer review and any attached files.

Reviewer #1: No

---

## [Editor Report · Acceptance letter]

24 Feb 2021

PONE-D-20-23871R1 

Immunoglobulin Replacement Therapy is critical and cost-effective in increasing life expectancy and quality of life in patients suffering from Common Variable Immunodeficiency Disorders (CVID): A health-economic assessment 

Dear Dr. de Vries:

I'm pleased to inform you that your manuscript has been deemed suitable for publication in PLOS ONE. Congratulations! Your manuscript is now with our production department. 

Kind regards, 

on behalf of

Professor Kevin Lu 

Academic Editor

PLOS ONE